# Co-production of an educational package for the universal human papillomavirus (HPV) vaccination programme tailored for schools with low uptake: a participatory study protocol

Harriet Fisher [1] ,[1] Suzanne Audrey,[1] Tracey Chantler [1] ,[2] Adam Finn,[3] Louise Letley,[4] Sandra Mounier-Jack,[2] Clare Thomas,[1] Julie Yates,[5] Matthew Hickman [1] [1]

For numbered affiliations see end of article.

**Correspondence to**
Dr Harriet Fisher;
Harriet.Fisher@bristol.ac.uk

## ABSTRACT

**Aim** To co-produce with young people an educational package about the human papillomavirus (HPV) vaccine that is tailored to increase vaccine uptake in schools and populations with lower uptake.

**Introduction** Persistent infection with HPV can result in cancers affecting men and especially women. From September 2019, the English-schools-based HPV vaccination programme was expanded to include young men (in addition to young women) aged 12–13 years. Some young people attending schools with lower uptake of the vaccine have unmet information needs. We hypothesise that mechanisms to address information needs and increase young people's autonomy in consent procedures will result in higher uptake.

**Methods and analysis** The Medical Research Council's framework for development and evaluation of complex interventions will inform intervention development. Recruitment of young people aged 12–15 years and key stakeholders (National Health Service commissioners, school staff, immunisation nurses and youth workers/practitioners) will be facilitated through existing links with healthcare organisations, schools and youth organisations in areas with lower uptake of the HPV vaccination programme. The proposed research will comprise three phases: (1) a rapid review of adolescent immunisation materials and preliminary qualitative interviews with young people and key stakeholders, (2) theory development and co-production of HPV vaccine communication materials through an iterative process with young people and (iii) testing delivery mechanisms and acceptability of the educational package in four schools with lower uptake.

**Ethics and dissemination** The University of Bristol's Faculty of Health Sciences and London School of Hygiene and Tropical Medicine's Research Ethics Committees provided approvals for the study. A dissemination event for young people and key stakeholders and webinar with the National Immunisation Network will be organised. The study findings will be published in peer-reviewed journals

### Strengths and limitations of this study

► This study aims to co-produce with young people an educational package that addresses young people's information needs about the human papillomavirus vaccination programme.

► The protocol for this study documents the use of an innovative, participatory methodological approach.

► Communication materials will be iteratively designed by the co-production team, taking into account young people's views at all stages of the development cycle.

► Findings from this study will be used to make recommendations for a future larger scale study.

► Parents, who also have unmet information needs, are not included in this study.

and presented at conferences. Recommendations for a future larger scale study will be made.

## INTRODUCTION

The human papillomavirus (HPV) vaccine currently used in England protects against infection from high-risk HPV types 16 and 18, which cause cancers affecting the cervix, vulva, vagina, penis, anus and oral cavity. The vaccine also protects against types 6 and 11, which cause 90% of genital warts. High coverage of the English HPV vaccination programme for young women aged 12–13 years has been achieved. Recent evidence for population-level effectiveness has highlighted the potential for HPV vaccination programmes to eradicate cervical cancer.[1 2]

However, national data often conceal within-country inequalities in uptake and access. Without concerted efforts to address

lower uptake among some populations, existing disparities in the incidence of cervical cancer by ethnicity and socioeconomic deprivation may increase.[3–5] Wide variations in uptake of the HPV vaccination programme across local authorities exist (range: 70.2%–95.8% for the first dose in 2018/2019).[6] We previously identified lower uptake by area and among some population groups, including minority ethnic groups.[7] Our research in schools with lower uptake showed complex sociocultural factors influence beliefs and priorities for young women to be vaccinated.[8] The requirement for written parental consent also acts as a major barrier,[9] which can be overcome to some extent by enabling verbal parental consent and adolescent self-consent.[10 11]

Based on emerging evidence, the Joint Committee on Vaccination and Immunisation concluded that extending the HPV vaccination programme to include young men could be cost effective.[12] The Department of Health announced the inclusion of young men from the 2019/2020 programme year. If sufficient uptake is attained, a universal vaccination programme will strengthen England's position to eliminate some HPV types and HPV-related cancers in the longer term. Men who have sex with men will also receive optimal protection by immunisation ahead of sexual debut.[12 13]

It is currently unknown whether the universal HPV vaccination programme will be delivered equitably by gender, or whether existing inequalities in uptake and incidence of HPV-related disease among some groups will be decreased or widened.[14] Inequalities may prevent delivery of the programme as intended, with insufficient understanding and engagement of young people and their parents or carers.

Despite largely positive public attitudes to vaccination,[15] recent calls have been made to tackle negative misconceptions of vaccines and limit health misinformation circulating in social media.[16] Many young people have limited or no understanding of the vaccines they receive or the diseases they are intended to prevent.[17]

Our previous research confirms some young women attending schools with lower uptake have unmet information needs, which can have implications for obtaining consent and vaccination uptake[8] (H Fisher *et al*, 2020, unpublished data). A systematic review, including two UK-based studies, showed young men have lower knowledge and understanding of HPV than their female counterparts.[18]

Methods to increase understanding, suggested by families and professionals, include: healthcare professional advocacy, accessible information formats, and delivery of educational sessions, in addition to the routine provision of information leaflets (H Fisher *et al*, 2020, unpublished data). Preliminary feedback from young people suggests the delivery of interactive HPV vaccine awareness raising sessions by a researcher in the school setting are acceptable. Young people's questions during the sessions related both to the HPV vaccine and preparation for the vaccination session.

A systematic review suggested weak evidence for the effectiveness of educational interventions at increasing acceptance, or intentions for their daughter to receive the HPV vaccine.[19] However, the review requires updating as the primary studies were predominantly related to hypothetical scenarios prior to vaccine licensing, and more studies have since been published.

Internationally, educational interventions about the HPV vaccine have been developed through research. These have targeted young adults,[20 21] parents[22–24] and professionals.[22–24] A complex intervention delivered in a USA's healthcare setting, comprising an educational component delivered by healthcare providers to young people aged 11–21 years, was shown to be effective at increasing uptake.[25] Similarly, a schools-based educational programme delivered through 113 face-to-face classes with Swedish students aged 16 years was shown to increase self-reported uptake of the HPV vaccine.[26] In Australia, a complex, schools-based intervention was not shown to be effective at increasing vaccination uptake, but improved adolescent knowledge and psychosocial outcomes.[27]

These studies suggest that an educational-based intervention to increase uptake has potential to be an acceptable and effective intervention in the context of the English universal HPV vaccination programme. We hypothesise that mechanisms to address information needs and increase young people's autonomy in consent procedures will result in higher uptake of the programme. We are unaware of any other educational packages that have been developed specifically for areas and populations groups with lower uptake,[7] or that have tested these specific mechanisms to increase uptake.

The aim of the study is to co-produce jointly with young people an HPV vaccination educational package that can be used within the adolescent schools-based immunisation programme and is tailored at increasing vaccine uptake in areas and populations with lower HPV vaccination coverage.

## METHODS AND ANALYSIS

We propose an innovative, participatory research approach involving co-production of an education package about the HPV vaccine with young people. The 2008 Medical Research Council's guidance for complex intervention development and evaluation will be followed.[28]

### Patient and public involvement

The design of this study was informed by our qualitative research undertaken with young women, parents, immunisation nurses and school staff[29] (H Fisher *et al*, 2020, unpublished data). We have consulted with the Bristol Young People's Advisory Group (YPAG) and Bristol City Youth Council about the design of the study. The Bristol YPAG comprises young people aged 10–17 years who are interested in healthcare and research. The Bristol City Youth Council is an elected group of 28 young people

from across the city. Both groups have been consulted about the design of the study and participant materials. They will also be invited to an event at the end of the study to consider findings and recommendations with the young people, healthcare professionals and school staff involved in the study.

The study will comprise three phases: (1) a rapid review of adolescent immunisation information materials and preliminary qualitative interviews with young people and key stakeholders, (2) theory development and co-production of HPV vaccine communication materials in an iterative process with young people and (3) examining delivery mechanisms and acceptability of the educational package in four schools with lower uptake of the HPV vaccination programme.

## Recruitment

During stages one and two, young men and women aged 12–15 years will be recruited from two youth community organisations in the south west of England. These organisations work with young people with similar sociodemographic backgrounds to the participants of our previous studies undertaken in schools with lower uptake. An additional youth community organisation in London has also been recruited. The age group 12–15 years is selected, as these young people will have diverse experiences in relation to the HPV vaccination programme. Key stakeholders will initially be identified through the research teams' existing relationships with youth organisations, immunisation teams and schools in the study areas. Additional interviews may be undertaken if other key individuals are identified as the activities progress.

## Stage one

A scoping review of existing young people's immunisation information materials will be undertaken. Web-based searches will be undertaken to locate online HPV vaccine communication tools targeted at young people in English-speaking countries. A description of the characteristics, format and content of the communication tools will be provided. The findings will be used to make recommendations to inform the content of the initial workshop.

Our qualitative research findings suggest that young people value external providers delivering health-related information in school settings. Preliminary individual or paired/peer group interviews (depending on the preference of participants) with young people (n~6) and key stakeholders (commissioners, school staff, immunisation nurses and youth workers/practitioners) (n~6) will be undertaken to explore further who should deliver the educational package and which key messages to cover.

## Stage two

Drawing on our previous qualitative research findings, it is envisaged that the educational package will include question and answer sessions alongside a series of short videos. Information needs may differ between schools depending on the sociodemographic characteristics of the student population. Therefore, additional videos will be co-produced to allow the educational session to be tailored as appropriate (eg, provision of information in multiple languages and additional information for parents on vaccine safety).

The materials will be co-produced by young people (n~6) and creative designers who are experienced in working with young people to create health-related media materials.

An initial workshop with young people (n~6–8) will be organised at a community youth organisation where they will be asked to (1) review existing communication materials; (2) comment on their understanding of key HPV vaccine messages; (3) make suggestions of their preferred messages, design and language style and (4) propose the type of media to be used (eg, animation and vloggers). Platforms to distribute the communication materials (eg, videos and powerpoints), which are accessible by young people, will be explored. Advice will also be sought from voluntary organisations, immunisation teams and school staff.

At least three additional workshops will be organised in different community youth organisations. Here, the materials will be iteratively redesigned by the co-production team. Young people's views will be taken into account at all stages of the development cycle.

## Stage three

Four schools with lower uptake of the HPV vaccination programme will be recruited to the study. Identification of potential schools to approach will be made in collaboration with the local immunisation teams in Bristol and London and our existing relationships with schools. It is envisaged that the educational package will last approximately 1 hour and be delivered to groups of vaccine eligible young people (aged 12–13 years). The package will be piloted approximately 2 weeks ahead of the scheduled vaccination session. Methods for obtaining young people's feedback will be explored (eg, short paper-based questionnaires and interactive quiz). Following the question and answer session, young people will be provided with consent forms and promotional materials available in different languages. Young people will be encouraged to share these with their parents/carers and return the completed consent form.

The feasibility of obtaining information from immunisation teams relating to HPV vaccine uptake by gender, ethnicity, deprivation and types of consent routinely obtained (no response, parent refusal, written or verbal parental consent, young person refusal or self-consent) will be assessed in terms of timeliness of data acquisition and levels of completeness for a future study.

A purposive sample of young people (n~12) from participating schools and key stakeholders (n~12) will be invited to take part in semi-structured or paired/peer group interviews (depending on the preference of participants) to gather information related to intervention acceptability, understanding of the HPV vaccine,

and perceptions of adolescent autonomy during consent. Potential key stakeholders to invite to interview will be identified by the research team as the research activities progress. The feasibility and sustainability of who delivers the education package will be further explored.

## Analysis

All interview recordings will be transcribed verbatim and any potentially identifying information removed. Familiarisation with the data will involve two researchers reading and discussing the transcripts to compare and begin to code and categorise the data. Thematic analysis[30] will be undertaken, within which similarities and differences will be explored. Data analysis will be assisted by the QSR NVivo V.12 software package.

Together, the findings from the rapid review and qualitative research will inform the logic model and development of theory for the educational package.

The outcome from this participatory research study will be a tailored educational intervention about the universal HPV vaccination programme. We will seek additional funding to test whether the educational package can: (1) improve uptake of the vaccination programme in schools/areas with lower uptake; (2) reduce inequalities in programme delivery; (3) increase young people's understanding about the HPV vaccine and (4) increase young people's autonomy in decision-making and consent procedures. For this larger scale study, we plan to design an intervention based on behaviour change theory (eg, COM-B model) which uses delivery of the educational materials we will have co-produced alongside other strategies to support behaviour change.

## ETHICS AND DISSEMINATION

In March 2020, the University of Bristol's Faculty of Health Sciences and London School of Hygiene and Tropical Medicine's Research Ethics Committees provided ethical approvals for the study (references: 99102 & 21887). The project began recruitment and data collection in April 2020 and the study's activities will finish in March 2021.

We will disseminate this research to the academic community, health and youth work practitioners and health policy workers. For the academic community, we plan to submit at least one peer-reviewed paper. We will also disseminate the results to at least one national and one international conference. For practitioners, we will complete a report of our findings and will disseminate the results through practitioner conferences and events.

We will also hold a community stakeholder event to which we will invite school nurses, school staff, parents and young people. This will be an opportunity to discuss the educational package for the HPV vaccine. Healthcare professionals will be actively encouraged to be involved in the dissemination activities, and findings from the study will also be disseminated through appropriate websites and social media. The final report of the study will also provide an opportunity for further dissemination.

**Author affiliations**
$^1$Population Health Sciences, Bristol Medical School, University of Bristol, Bristol, UK
$^2$Department of Global Health and Development, London School of Hygiene & Tropical Medicine, London, UK
$^3$Division of Clinical Sciences, University of Bristol, Bristol, UK
$^4$National Infection Service, Public Health England, London, UK
$^5$Public Health England, Taunton, UK

**Acknowledgements** The study is supported by the National Institute for Health Research (NIHR) Health Protection Research Unit in Behavioural Science and Evaluation at the University of Bristol. The views expressed are those of the authors and not necessarily those of the National Health Service, the NIHR, the Department of Health or Public Health England. This study is supported by the NIHR's Health Protection Research Unit in Vaccines and Immunisation, a partnership between Public Health England and the London School of Hygiene & Tropical Medicine.

**Contributors** All authors were involved in the conception and design of the research. MH is the principal Investigator and contributes expertise in researching infectious disease control. SA contributes expertise in qualitative research and process evaluation. TC provides implementation, mixed methods and anthropology research expertise in relation to UK vaccination programmes. AF provides experience in the field of vaccine and policy development. LL provides expertise in implementing the national human papillomavirus vaccination programme. SM-J has expertise in health systems and policy evaluations, notably vaccination programmes. CT offers expertise in knowledge mobilisation. JY will support the dissemination of findings and wider implementation of the educational package as appropriate. As study manager, HF will be responsible for coordinating the study and provides links with community organisations and schools. HF wrote the first draft and all authors contributed to the final version of the manuscript.

**Funding** This work is supported by the Medical Research Council's Public Health Intervention Development scheme (project number: MR/T027150/1).

**Competing interests** None declared.

**Patient and public involvement** Patients and/or the public were involved in the design, or conduct, or reporting, or dissemination plans of this research. Refer to the Methods section for further details.

**Patient consent for publication** Not required.

**Provenance and peer review** Not commissioned; externally peer reviewed.

**ORCID iDs**
Harriet Fisher http://orcid.org/0000-0002-5639-0955
Tracey Chantler http://orcid.org/0000-0001-7776-7339
Matthew Hickman http://orcid.org/0000-0001-9864-459X

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
