## [Reviewer comments · BMJ Open]

ARTICLE DETAILS

TITLE (PROVISIONAL)	Co-production of an educational package for the universal human papillomavirus (HPV) vaccination programme tailored for schools with low uptake: A participatory study protocol
AUTHORS	Fisher, Harriet; Audrey, Suzanne; Chantler, Tracey; Finn, Adam; Letley, Louise; Mounier-Jack, Sandra; Thomas, Clare; Yates, Julie; Hickman, Matthew

VERSION 1 – REVIEW

REVIEWER	Kristin Oliver Icahn School of Medicine at Mount Sinai USA
REVIEW RETURNED	08-Jul-2020

GENERAL COMMENTS	This is an important topic. There is a clear need for strategies to improve HPV vaccination in the school setting, and currently a limited number of tools directed at adolescents. A clearly written and well organized protocol.
--

REVIEWER	Alice Forster UCL, UK
REVIEW RETURNED	24-Jul-2020

GENERAL COMMENTS	This is a well written study protocol, describing a series of projects to co-produce an educational package to inform young people about the HPV vaccine, with the intention of it increasing HPV vaccine uptake. I have two main comments that I would like to see further discussion / clarification on: the choice of intervention, and additional detail regarding the method. I look forward to seeing the study results. I am personally sceptical that educational interventions can result in improvements in vaccine uptake on their own, although I appreciate that this educational intervention would be delivered in the context of self-consent procedures. I am not suggesting that the authors should not conduct this study, but I think this protocol could better situate the reader in the current evidence supporting the use of educational interventions. It would be useful to include a discussion of reviews that have been done in this area - for example, Fu et al. 2014 Vaccine - which found that there are few well conducted education intervention studies that find improvements in vaccine uptake. Can the authors explain why their educational intervention is going to be different / better / result in a different outcome than those that have gone before it? Related to this discussion:
---

	Reference 23 - the intervention was conducted with health professionals, not young people as described; Reference 24 - it may be worth noting how labour intensive the intervention was. The detail of the proposed methodology is somewhat limited: - Stage One. Are the authors interested in information materials that have been used in the UK only? How will this information be identified? How will the young people and stakeholders be identified / recruited? - Stage three. How will the feasibility of data collection be assessed? How will schools be identified / recruited, as well as stakeholders? - Finally, what is the status of this study? When will it be conducted? Has data collection finished?
--	---

VERSION 1 – AUTHOR RESPONSE

Reviewer 1.

This is an important topic. There is a clear need for strategies to improve HPV vaccination in the school setting, and currently a limited number of tools directed at adolescents. A clearly written and well organized protocol.

Response: We thank the reviewer for their supportive comments.

Reviewer 2.

This is a well written study protocol, describing a series of projects to co-produce an educational package to inform young people about the HPV vaccine, with the intention of it increasing HPV vaccine uptake. I have two main comments that I would like to see further discussion / clarification on: the choice of intervention, and additional detail regarding the method. I look forward to seeing the study results.

Comment 1: I am personally sceptical that educational interventions can result in improvements in vaccine uptake on their own, although I appreciate that this educational intervention would be delivered in the context of self-consent procedures. I am not suggesting that the authors should not conduct this study, but I think this protocol could better situate the reader in the current evidence supporting the use of educational interventions. It would be useful to include a discussion of reviews that have been done in this area - for example, Fu et al. 2014 Vaccine - which found that there are few well conducted education intervention studies that find improvements in vaccine uptake. Can the authors explain why their educational intervention is going to be different / better / result in a different outcome than those that have gone before it?

Response 1: We wholeheartedly agree with the reviewer's comments that behavioural change, such as vaccination uptake, can be difficult to achieve through provision of information alone. This work is in response to feedback received from young people that they do not feel they have access to the relevant information. Further, we are focusing on schools where uptake has historically been low, with the aim of trying to understand if there are particular information needs in these schools and to tailor the education package accordingly. Whilst behaviour change theory shows that education alone is unlikely to result in change, ensuring young people have accessible and relevant information to support their choices is a key component. In addition to uptake of the HPV vaccination programme, we are also interested in establishing whether the educational package can improve young people's understanding and autonomy in consent procedures. We believe these are important outcomes in

their own right. This is described towards the end of the manuscript (page 10, paragraph 4).

For this research study we plan to develop communication materials and test delivery of an educational package in a small number of schools to establish the acceptability of the intervention and the feasibility of obtaining information relating to HPV vaccine uptake. We will only seek additional funding to test the effectiveness at increasing uptake if our study findings show the educational package is acceptable and feasible, and whether there is any evidence of promise in relation to changes to uptake. For a larger scale study, we plan to design an intervention based on behaviour change theory (e.g. COM-B model) which uses delivery of the educational materials we will have co-produced alongside other strategies to support behaviour change. This has also been clarified in the manuscript (page 10, paragraph 4).

We are aware of the systematic review that was undertaken by Fu et al. However, the majority of primary studies comprising the review related to hypothetical scenarios and do not report impact on uptake of HPV vaccination programmes. For this reason we did not originally include a reference to it. Instead, we provide detail on individual studies reporting educational interventions where impact on uptake of the HPV vaccine was reported. As described in the manuscript, these educational-based interventions show promise for an acceptable and effective intervention in the context of the English HPV vaccination programme.

Page 5, paragraph 4: We have incorporated the following sentence within the introduction section so the reader is aware of this systematic review by Fu et al: 'A systematic review suggested weak evidence for the effectiveness of educational interventions at increasing acceptance, or intentions for their daughter to receive the HPV vaccine [21]. However, the review requires updating as the primary studies were predominantly related to hypothetical scenarios prior to vaccine licensing, and more studies have since been published.'

Comment 2: Related to this discussion: Reference 23 - the intervention was conducted with health professionals, not young people as described

Response 2: Page 6, paragraph 1. We have amended the sentence as follows: 'A complex intervention delivered in a United States of America healthcare setting, comprising an educational component delivered by healthcare providers to young people aged 11 to 21 years, was shown to be effective at increasing uptake'.

Comment 3: Reference 24 - it may be worth noting how labour intensive the intervention was.

Response 3: Page 6, paragraph 1. We have amended the sentence as follows: Similarly, a schools-based educational programme delivered through 113 face-to-face classes with Swedish students aged 16 years was shown to increase self-reported uptake of the HPV vaccine

Comment 4: The detail of the proposed methodology is somewhat limited:

- Stage One. Are the authors interested in information materials that have been used in the UK only? How will this information be identified? How will the young people and stakeholders be identified / recruited?

Response 4: Page 7, paragraph 3. We have clarified that recruitment of young people during Stages One and Two of the study will be facilitated by three youth organisations who have agreed to participate in the study. Further, we clarify key stakeholders will initially be identified through the research teams existing relationships with youth organisations, immunisation teams and schools in the study areas. Additional interviews may be undertaken if other key individuals are identified as the activities progress.

Page 7, paragraph 4. We have provided the following information to provide clarity in relation to the purpose and methodology for the scoping review: 'A Web-based searches will be undertaken to locate online HPV vaccine communication tools targeted at young people in English speaking countries. A description of the characteristics, format, and content of the communication tools will be provided. The findings will be used to make recommendations to inform the content of the initial workshop.'

Comment 5: Stage three. How will the feasibility of data collection be assessed? How will schools be identified / recruited, as well as stakeholders?

Response 5: Page 9, paragraph 4. We have clarified the assessment of acquiring HPV vaccine uptake data by amending the following sentence: 'The feasibility of obtaining information from immunisation teams relating to HPV vaccine uptake by gender, ethnicity, deprivation, and types of consent routinely obtained (no response, parent refusal, written or verbal parental consent, young person refusal or self-consent) will be assessed in terms of timeliness of data acquisition and levels of completeness for a future study.'

Page 9, paragraph 3. We have clarified the following in relation to the identification and recruitment of schools: 'Identification of potential schools to approach will be made in collaboration with the local immunisation teams in Bristol and London and using our existing relationships with schools.'

Page 9, paragraph 5. We have clarified our key stakeholders will be identified by the research team as the study progresses through the following sentence: 'Potential key stakeholders to invite to interview will be identified by the research team as the research activities progress.'

Comment 5: Finally, what is the status of this study? When will it be conducted? Has data collection finished?

Response 5: Page 10, paragraph 5. The following has been clarified: 'The project began recruitment and data collection in April 2020 and the study activities will finish in March 2021.'

VERSION 2 – REVIEW

REVIEWER	Alice Forster UCL, UK
REVIEW RETURNED	18-Sep-2020
GENERAL COMMENTS	This protocol describes an interesting study, and I look forward to reading the results. Thank you to the authors for addressing my comments; I have no further comments.